# Anti-Cancer Activity of Verteporfin in Cholangiocarcinoma

**DOI:** 10.3390/cancers15092454

**Published:** 2023-04-25

**Authors:** Jihye L. Golino, Xin Wang, Jing Bian, Benjamin Ruf, Michael Kelly, Baktiar O. Karim, Maggie C. Cam, Changqing Xie

**Affiliations:** 1Thoracic and GI Malignancies Branch, Center for Cancer Research, National Cancer Institute, National Institutes of Health, Bethesda, MD 20892, USA; 2CCR Collaborative Bioinformatics Resource, National Cancer Institute, National Institutes of Health, Bethesda, MD 20892, USA; 3Frederick National Laboratory for Cancer Research, Leidos Biomedical Research, Inc., Frederick, MD 21701, USA; 4Molecular Histopathology Laboratory, Leidos Biomedical Research, Inc., Frederick, MD 21701, USA; 5Liver Cancer Program, Center for Cancer Research, National Cancer Institute, National Institutes of Health, Bethesda, MD 20892, USA

**Keywords:** cholangiocarcinoma, verteporfin, cancer stem cells, YAP

## Abstract

**Simple Summary:**

Cholangiocarcinoma (CCA) is a highly lethal malignancy, and its prognosis is poor. There are unmet needs to develop effective therapies. The overexpression of Hippo/YAP pathway and the association of Hippo/YAP pathway with an immunosuppressive microenvironment is indicated with bulk RNA sequencing data. In this study, we investigated the antitumoral effect of verteporfin in CCA YAP/AKT murine models. We found that verteporfin reduced liver weight and tumor formation in CCA YAP/AKT mice. Our results also showed the change in immune cell composition in liver/tumors with the treatment of verteporfin as well as the inhibition of cancer stemness. Our data suggest the potential application of verteporfin in patients with an overexpression of Hippo/YAP pathway.

**Abstract:**

Cholangiocarcinoma (CCA) is a heterogenous malignancy that arises from the biliary epithelium and has a poor clinical prognosis. The Hippo/yes-associated protein (YAP) pathway has been reported to affect various aspects of tumorigenesis, with high expression of YAP1 being negatively associated with survival in CCA patients. Thus, we investigated the antitumoral effect of verteporfin, a YAP1 pathway inhibitor, in YAP1/AKT hydrodynamic tail vein injected murine models. We also used flow cytometry and single-cell RNA sequencing (scRNA-seq) to analyze the change in the immune cell profile and malignant cell stemness following verteporfin treatment. Our results demonstrated reduced liver weight and tumor formation in verteporfin-treated groups compared to that of a vehicle-treated group. Immune cell profiling through flow cytometry showed that relative to the vehicle, verteporfin induced a higher ratio of tumor-associated macrophage (TAM) M1/M2 and increased the percentage of activated CD8 T cell population (CD8+CD25+ and CD8+CD69+). scRNA-seq analysis showed significantly increased TAM M1 populations following verteporfin treatment and decreased proportions of stem-like cells within the malignant cell population. In summary, this study indicates that in CCA YAP/AKT murine models, verteporfin reduces tumorigenesis by polarizing anti-tumoral TAM and activating CD8 T cells and decreasing stem-like malignant cell proportions in the tumor microenvironment.

## 1. Background

Cholangiocarcinoma (CCA) is a heterogeneous malignancy originating from the biliary epithelium in the biliary tree system that has become one of the leading causes of liver cancer-related deaths worldwide. The occurrence of CCA has consistently increased across sex/race/ethnic populations [1,2] and is often diagnosed too late due to nonspecific symptoms. For these reasons, CCA has a poor prognosis with a low five-year survival rate [3] despite the recent advances in chemotherapy [4,5,6], targeted therapies [7,8,9] and the combination of immunotherapy and chemotherapy [10,11]. This underscores the demand for the development of novel effective treatment strategies for CCA patients, especially at its advanced stage.

Yes-associated protein (YAP) is a downstream transcriptional regulator of the Hippo signaling pathway, which activates multiple oncogenic pathways/target genes and interacts with other oncogenic proteins/signaling pathways in numerous types of cancers. Signal transducer and activator of transcription 3 (STAT3)-YAP/TAZ (transcriptional coactivator with PDZ-binding motif) signaling also promotes angiogenesis [12] and is associated with poor survival in breast cancer [13]. In non-small cell lung cancer, the overexpression of YAP induces proliferation, while depletion of YAP causes growth arrest [14]. Moreover, YAP/TAZ promotes hepatocyte proliferation and tumor growth upon Myc-β-catenin in hepatocellular carcinoma (HCC) [15] and mediates sorafenib resistance in HCC [16]. In CCA, YAP maintains cancer stemness and its expression is negatively correlated with survival in iCCA patents [17,18]. In mice models, the overexpression of YAP1 along with AKT triggers CCA formation [19], which suggests that YAP1 pathway is a potential target to control CCA.

Verteporfin is a drug that was originally developed and used as a photosensitizer for photodynamic reaction. Verteporfin inhibits the overgrowth of the liver induced by YAP overexpression [20] and upregulation of cancer stemness markers [18]. We recently showed that verteporfin reduces CCA cell growth but enhances cell apoptosis in a dose-dependent manner. Nevertheless, verteporfin was shown to inhibit stemness in vitro and synergize the anti-tumoral efficacy of anti-PD-1 in a xenograft subcutaneous CCA murine model [21]. In this study, we tested the efficacy of verteporfin in a YAP/AKT CCA mouse model, established through the hydrodynamic tail vein injection of plasmids of YAP1 and AKT. We investigated immune cell/cancer stemness changes induced by verteporfin treatment using flow cytometry and single-cell RNA sequence (scRNA-seq) analysis.

## 2. Methods

### 2.1. YAP/AKT CCA Mice Model and Tissue Process

Six- to eight-week-old C57BL/6 mice were purchased from Charles River Laboratory (Wilmington, MA, USA). The mice underwent hydrodynamic tail vein injections with the plasmid mixture as previously described [22], consisting of 30 µg of YAP1, 20 µg of AKT, and 2 µg of HSB2 plasmids. The mixture was dissolved in a total volume of 1600 µL of PBS. The plasmids were prepared by growing in E. coli cultures and isolated using a Plasmid DNA Maxiprep Kit (MACHEREY-NAGEL, Düren, Germany). Plasmid concentrations were measured using a NanoDrop Spectrophotometer^TM^ (Thermo Fisher Scientific, Frederick, MD, USA). Mice were euthanized at 8 weeks after the plasmid injection and 5 weeks after the treatment. Sections of the mouse liver samples were collected and fixed in a formaldehyde solution. The fixed liver samples were trimmed, paraffin blocks and slides were made, and hematoxylin and eosin stain (H&E) staining was performed by Histoserv (Germantown, MD, USA). Quantification of the stained area was observed under Halo software in the Molecular Histopathology Laboratory (MHL) of the National Cancer Institute (NCI) (Frederick, MD, USA). Hematoxylin and eosin-stained sections were scanned at 20× objective magnification (0.5 μm/pixel) using an Aperio AT2 digital whole-slide scanner (Leica Biosystems, Deer Park, IL, USA). The presence of CCA and extent of tissue infiltration were confirmed by an experienced murine histopathologist. The remaining liver samples were processed for flow cytometry and scRNA-seq analysis as mentioned below. All experiments were conducted according to the local institution guidelines and approved by the Animal Care and Use Committee of the National Institutes of Health (Bethesda, MD, USA).

### 2.2. Verteporfin In Vivo Treatment

Three weeks after the tail vein injection, mice were randomly split into two groups treated with either vehicle (DMSO, containing PBS solution) or verteporfin (in PBS, 100 mg/kg) (Millipore Sigma, St. Louis, MO, USA) every three days for 5 weeks, respectively (Appendix A). Eight weeks after the plasmid injection and 5 weeks after the initial treatment, all mice were euthanized using carbon dioxide asphyxiation. Mouse livers were removed and washed. The weight of the livers was measured and compared between the groups as the indication of tumorigenesis.

### 2.3. Flow Cytometry

Livers were removed immediately after the mice were sacrificed. After homogenization, debris was removed by nylon mesh filtering. Immune cells were isolated by isotonic Percoll centrifugation (850× *g*, 25 min, with full acceleration and without deceleration). The cells were incubated with indicated antibodies for 30 min at 4 °C for surface marker staining after red blood cells were lysed with ammonium-chloride-potassium lysing buffer. Dead cells were excluded by using a Zombie UV™ Fixable Viability Kit (Biolegend, San Diego, CA, USA). The following mouse antibodies were used for flow cytometry analysis: (1) for immunoprofiling: anti-Ly-6G-Alexa Fluor (AF) 700 (clone 1A8; Biolegend), anti-CD4-Brilliant Violet (BV) 605 (clone GK1.5; Biolegend), anti-mouse-Ly-6C-APC/Cyanine7 (clone HK1.4; Biolegend), anti-CD3-PE (clone 17A2; Biolegend), anti-CD19-PerCP/Cyanine5.5 (clone 1D3/CD19; Biolegend), anti-F4/80-FITC (clone BM8; Biolegend), anti-CD11b-Pacific Blue (PB) (clone M1/70; Biolegend), anti-I-A/I-E (MHC-II)-BV510 (clone M5/114.15.2; Biolegend), anti-CD8-BV786 (clone 53-6.7; BD Biosciences, Franklin Lakes, NJ, USA), anti-mouse CD11c (clone N418; Biolegend) (2) T cells activation/exhaustion: anti-CD45R/B220-AF700 (clone RA3-6B2; Biolegend), anti-F4/80-AF700 (clone BM8; Biolegend), anti-CD11b-AF700 (clone M1/70; Biolegend), anti-CD3-AF594 (clone 17A2; Biolegend), anti-CD4-BV605 (clone GK1.5; Biolegend), anti-CD8-BV786 (clone 53-6.7; BD Biosciences), anti-CD25-PerCP/Cyanine5.5 (clone 3C7; Biolegend), anti-CD279 (PD-1)-FITC (clone 29F.1A12; Biolegend), anti-CD39-PE/Cyanine7 (clone Duha59; Biolegend) anti-CD69-BV650 (clone H1.2F3; Biolegend) (3) myeloid cells: anti-F4/80-FITC (clone BM8; Biolegend), anti-Ly6C-AF700 (clone HK1.4; Biolegend), anti-CD86-APC/Cyanine7 (clone GL-1; Biolegend), anti-CD80-PE (clone 16-10A1; Biolegend), anti-CD206-BV605 (clone C068C2; Biolegend), anti-CD11b-PB (clone M1/70; Biolegend), anti-CD163-PE/Cyanine7 (clone S15049I; Biolegend), anti-mouse CD11c (clone N418; Biolegend). The frequency of each immune cells was calculated by the percentage of the total of each parent cell as indicated by the y-axis of each graph. All stained cells were analyzed using CytoFLEX LX platforms, and the results were analyzed using FlowJo software version 10.8 (BD).

### 2.4. Single Cell RNA Sequencing

Liver samples were processed using a mouse Tumor Dissociation Kit from Miltenyi Biotec (Bergisch Gladbach, North Rhine-Westphalia, Germany) following the manufacturer’s instructions. Liver tissues were cut into 2–4 mm pieces and transferred into a gentleMACS C Tube (Miltenyi #130-093-237), dissociated by gentleMACS Octo Dissociator (Miltenyi Biotec), and incubated at 37 °C for 40 min (200 rpm). After incubation, tissues were dissociated using a gentleMACS Octo Dissociator (Miltenyi Biotec) and filtered using 70 μm of MACS SmartStrainers (Miltenyi Biotec #130-098-462). Red blood cells and debris were removed by Red Blood Cell Lysis solution (Miltenyi Biotec #130-094-183) and Debris Removal Solution (Miltenyi Biotec #130-109-398), respectively. Cells were submitted to the CCR Single Cell Analysis Facility at NCI (Bethesda, MD, USA).

### 2.5. Library Preparation and Sequencing for Mouse Sample

Single-cell sequencing was performed using 10x Genomics scRNA-Seq 3′ v3.1 according to the manufacturer’s instructions. Cell suspensions were assessed and counted with acridine orange and propidium iodine fluorescence dye on an automated cell counter (LunaFL, Logos Biosystems) (Annandale, VA, USA) and adjusted for single-cell partitioning to target approximately 6000 datapoints per sample when possible. For single-cell library preparation, as defined in the 10x Genomics user guide, following cell partitioning with barcoded gel beads, the cells are lysed, and poly-adenylated transcripts are reverse-transcribed with the inclusion of a cell-specific barcode and a unique molecular identifier. Partitioning droplets are broken, and barcoded cDNA is amplified for 14 cycles before Illumina-based sequencing libraries are prepared by fragmenting cDNA and adding necessary sequencing adapters along with a sample-specific index barcode. For sample preparation on the 10x Genomics platform, the Chromium Next GEM Single Cell 3′ Kit v3.1 (PN-1000268), Chromium Next GEM Chip G (PN-1000120) and Dual Index Kit TT Set A (PN-1000215) were used. The molarity of each library was calculated based on the concentration and library size measured using a Bioanalyzer (Agilent Technologies, Santa Clara, CA, USA). Libraries were pooled and normalized to a final loading concentration. The sequencing run was set up as recommended with 28 cycles + 10 cycles + 10 cycles + 90 cycles. Demultiplexing was performed using the cellranger mkfastq pipeline, which allows for one mismatch in the sample index barcodes. Raw reads were aligned to the mm10 reference genome (refdata-gex-mm10-2020A) to generate a per-cell gene expression count matrix with cellranger count (cellranger v6.1.2, 10x Genomics). A per-cell mean sequencing depth of 50,000 reads/cell was targeted for each sample. Libraries were sequenced on an Illumina NextSeq 2000.

### 2.6. Murine CCA scRNA-Seq Data Analysis

Filtered feature-barcode matrix.h5 files from cellranger output for all samples were merged into a Seurat Object using the Seurat workflow [23]. Cells were preprocessed using Unique Molecular Identifier (UMI) counts, the number of expressed genes, and mitochondrial content; cells with low UMI counts (>500) or low complexity (<0.5 genes/UMI) were filtered from the data, along with cells whose gene or mitochondrial content exceeded 3 absolute deviations above the respective medians. The gene expression data were then normalized using the Seurat SCTransform function [23]. Downstream analyses involving differential gene expression (DEG) and gene set enrichment analyses (GSEA) were performed within the NIH Integrated Analysis Portal (NIDAP) using R programs developed on the Palantir Foundry platform (Palantir Technologies, Washington, DC, USA). All scRNA-seq data were submitted to the Gene Expression Omnibus (GEO) public database at the NCBI. The code used for the analysis was deposited in GitHub (https://github.com/NIDAP-Community/Anti-cancer-activity-of-verteporfin-in-cholangiocarcinoma, accessed on 20 February 2023). Raw data were deposited in GEO (GSE229855).

Highly variable genes were outlined by principal component analysis (PCA) and the first 15 principal components were further projected as Uniform Manifold Approximation and Projection (UMAP) plots [24]. The number of principal components to be used was calculated using the Elbow method. Unsupervised clustering was achieved using the Seurat FindClusters function [23]. Cell clusters were illustrated according to the DEG and canonical marker genes. Sub-clustering analysis was conducted by re-running FindClusters on filtered subsets.

### 2.7. Cell Identification

For each individual cell, the average expression of immune cell markers in previously published literature [25] was calculated using the Seurat AddModuleScore function. Cells were then defined based on the marker set with the highest average. The copy number variation (CNV) across epithelial cells was calculated using inferCNV [26]. A cutoff of 0.1 was used to screen cells possessing low gene counts and an sd_amplifier value of 2 was applied to account for background noise. A copy number score (CNS) was arranged for each cell as the formula below:$$CNS=\Sigma {\left(CNV_{gene}-mean\left(CNV\right)\right)}^{2}$$

Epithelial cells with a CNV in the top 25 percentile were further classified as malignant cells. The remaining epithelial cells were defined as cholangiocytes.

### 2.8. Differential Expression Analysis and Gene Set Variation Analysis (GSVA)

Differential gene expression analysis was performed on log-normalized data using the limma function and according to the pseudobulk approach outlined in [23]. GSEA using the fgsea (version 1.8.0) R package was then run on the ranked list of differentially expressed genes. Pathways coinciding with important gene sets were referred from the H:Hallmark, CP:KEGG, and CP:Reactome collections within the Molecular Signature Database (MSigDB) (v2022.1.Mm). Pathways characterizing similar biological functions were eliminated from the visualization.

### 2.9. Human Bulk Transcriptomic Analysis of CCA Samples

TIMER2.0 was used to compare the expression level of the YAP signaling gene signature between non-tumor and tumor tissues and analyze the correlation between YAP1 signaling expression and stemness marker gene expression and infiltrating immune cells (http://timer.comp-genomics.org/, accessed on 20 February 2023) [27].

### 2.10. Statistical Analysis

Statistical analysis was performed with GraphPad Prism 8 (GraphPad Software). The significance of the difference between groups was calculated by Student’s unpaired *t* test. *p* < 0.05 was considered statistically significant.

## 3. Result

### 3.1. YAP1 Pathway Correlated with Cancer Stemness and Stromal Cells in CCA

To compare the expression levels of YAP1 signaling pathway genes between tumor and normal tissue samples, we used the TIMER2.0 platform to evaluate the expression profiles based on the cholangiocarcinoma cohort (CHOL) obtained from TCGA (The Cancer Genome Atlas) database [27]. As shown in Figure 1A and Appendix A, we found that the expression levels of YAP1, Transcriptional enhanced the associated domain (TEAD) family and YAP signaling signature in tumor tissues of CCA and were significantly higher than the corresponding normal tissues. There is a positive correlation of the expression level between the YAP1 pathway gene signature and key stemness transcription factor SOX9 (Figure 1B and Appendix A). Moreover, the expression of the YAP1 pathway gene signature was significantly and positively associated with the infiltrating levels of immunosuppressive stromal cells, including cancer-associated fibroblasts, macrophage and Tregs, but negatively associated with antitumoral CD4 Th1 (Figure 1C and Appendix A). These results indicate that the YAP1 pathway plays profound roles related with CCA stemness and immunosuppressive tumor microenvironment (TME).

### 3.2. Antitumoral Efficacy of Verteporfin in YAP/AKT Mouse Model

Since the liver weight represents the overall tumor growth or formation in this model, the liver weight was measured and compared between DMSO (vehicle-treated group) and verteporfin-treated groups (Figure 2A,B). As a result, the liver weights of verteporfin-treated mice were significantly reduced (21%) compared to the liver weights of vehicle-treated mice (Figure 2A, *p* = 0.0375). The tumor area was quantified using H&E staining (Figure 2C). The stained tumor area was calculated based on the whole liver area as a percentage (%). The tumor areas of verteporfin-treated mice were significantly lower in comparison to the vehicle-treated group (Figure 2D, *p* = 0.0097). Therefore, verteporfin reduced tumorigenesis in the YAP/AKT mouse model.

### 3.3. Verteporfin Treatment Modulates Immune Cell Landscape of CCA in YAP/AKT Mouse Model

To investigate whether verteporfin modulates the TME that leads to the control of CCA growth in the YAP/AKT mouse CCA model, immune cells from whole liver tissues were isolated and analyzed using flow cytometry as described in the Materials and Methods section. B cells, CD4 and CD8 T cells, CD11b+ myeloid cells, macrophages and dendritic cells were identified (Appendix A). Gating strategies were shown in Appendix A. Although there was no significant change in the percentage of CD8 T cells between the vehicle and verteporfin treatment groups (Appendix A), the proportion of activated CD8 T cells (CD25+CD8+) were significantly increased along with a decreased proportion of memory CD8 T cells (CD69+CD8+) (Figure 3A). While the proportion of individual exhausted CD8 T cells [CD39+CD8+ and programmed cell death protein-1 (PD-1)+CD8+] showed no change, PD-1+CD39+ cells from CD8 T cells were significantly reduced in verteporfin treated group (Figure 3B), which suggests that verteporfin may inhibit complete/terminal exhaustion of CD8 T cells. In addition, verteporfin increased the percentage of activated CD4 T cells (CD25+CD4+) and decreased the percentage of memory CD4 T cells (CD69+CD4+) (Figure 3C) but had no effect on the overall percentages of the CD4 T cell population and exhausted CD4 T cells (Figure 3D).

Macrophages play a critical role in tumorigenesis. To study this, we tested the changes in TAM-M1 and TAM-M2 macrophages (CD80 and CD86 for TAM-M1 and CD163 and CD206 for TAM-M2, respectively). The proportion of CD163+ macrophages was significantly reduced in the verteporfin-treated group (*p* < 0.0001), whereas there was a decrease in the proportions of CD86- and CD80-positive cell populations (Figure 4A). However, when looking at the relative proportions of TAM-M1 and TAM-M2 macrophages as measured by CD80/CD163 or CD86/CD163 ratios, verteporfin treatment was noted to significantly increase the TAM M1/M2 ratio (Figure 4B). Although there was an increased trend of proportions of B cells and dendritic cells observed (Appendix A) in the verteporfin-treated group, the changes were not significant.

To further illustrate how verteporfin affects TME dynamics in the YAP/AKT CCA mouse model, a total of 33,769 isolated single cells was obtained from mouse normal livers or tumors, which covered various tumorigenic stages of CCA. In addition, we performed scRNA-seq on livers or tumors from a YAP/AKT mouse model treated either with a vehicle or verteporfin. A total of 14 clearly separated cell clusters were identified (Figure 5A,B, Appendix A). Based on the expression of known markers, we identified endothelial cells, hepatocytes, epithelial cells, immune cells, and fibroblasts (Figure 5A, Appendix A). The immune cells were comprised of CD4, CD8 and Treg, NK cells, B cells and myeloid cells including dendritic cells, TAM-M1 and TAM-M2 (Figure 5A,B, Appendix A).

Remarkably, the proportion of TAM-M1 cells, which function as pro-inflammatory/anti-tumorigenic immune cells, was greatly induced (fivefold) by verteporfin treatment, while TAM-M2 did not show a significant difference (Figure 5B and Appendix A). The proportions of B cells and CD8 T cells decreased in the verteporfin-treated group, though these were insignificant changes (Appendix A). We further analyzed the subset of CD8, CD4 and TAMs (TAM-M1+TAM-M2) cell population based on the results from flow cytometry (Figure 3, Figure 4 and Figure 5). As shown in Appendix A, only the change in the percentage of CD80+ TAMs (TAM-M1) among TAMs significantly increased after verteporfin treatment. Other changes showed a similar, although not significant, tendency to the flow results, for example, CD25+CD8+ and PD1+CD39+CD8+ cell populations.

### 3.4. Verteporfin Reduced Expression of Cancer Stemness Genes in Malignant Cells

Overall, verteporfin significantly suppressed the Hippo/Yap1 pathway as determined through GSEA (Figure 6A and Appendix A), indicating the effectiveness of treatment. To further investigate how verteporfin affects malignant cells, we compared the transcriptomic changes in the malignant cell population between the verteporfin treatment and vehicle control groups. We found many significant changes in this population treated with verteporfin (Appendix A, Appendix A, Appendix A). GSEA analysis indicated upregulated interferon α and γ signaling pathways, inflammatory response, and adaptive/innate immune process with the treatment of verteporfin (Figure 6B, Appendix A), which were consistent with the antitumoral outcome of verteporfin. Verteporfin treatment downregulated biosynthesis of steroids, lipoproteins, and cholesterol (Figure 6C). These results indicated that verteporfin treatment induced an immune response within tumors as well as altering the liver function.

We further questioned whether the proportion of cells with positive cancer stem cell markers was affected by verteporfin treatment (Figure 6D). Overall, most of the proportions of stemness gene-expressing cells were reduced by verteporfin treatment (Figure 6E). Based on the results, verteporfin might inhibit tumorigenesis in CCA via down-regulation of cancer stemness.

## 4. Discussion

In this study, we investigated the anti-tumoral effect of verteporfin in the YAP/AKT mouse CCA model and changes in malignant cells and immune cell compartment. Consistent with the results in a previous study [21], verteporfin treatment reduced the liver weight and tumor area while modulating immune cell profiles and suppressing cancer stemness.

To the best of our knowledge, the effect of YAP inhibition on immune cell modulation has not been widely reported. T cell activation is a pivotal event in the adaptive immune response, which leads to the production and release of proinflammatory cytokines. Eventually, activated T cells interact with the antigens on their target cells and then results in cytotoxicity, apoptosis and cell destruction [28]. Here, we showed that verteporfin treatment induced activation markers of CD25 in CD4 and CD8+ T cells according to our flow cytometry result, supporting that modulation of the immune response mediated by T cell activation might be one of factors in verteporfin causing tumor suppression in the YAP/AKT CCA mouse model.

Early and late dysfunctional tumor-specific T cells can be characterized by surface marker expression. PD1 and lymphocyte-activation gene 3 (LAG3) is expressed during both early and late stages of dysfunctional T cells, but late dysfunctional T cells express additional inhibitory receptors, such as CD38, CD39, CD101 and TIM3 (T-cell immunoglobulin and mucin-domain containing 3) [29]. Our results show that verteporfin treatment reduced the proportions of double-positive PD1+CD39+CD8+ T cells. At an early stage of tumor development, T cells undergo an anergy-like early dysfunctional state that allows cancer cells to grow. Constant stimulation by tumor antigens with cancer progression triggers a late dysfunctional state. T cell exhaustion mechanisms may regulate the loss of cytotoxic effector function, including cytokines and/or cytotoxic molecules, such as interferon-γ [30]. Therefore, reduced PD1+CD39+ in CD8+ T cell populations mean that the anti-tumor effect of verteporfin might result from the inhibition of terminal exhaustion/dysfunctional CD8+ T cells.

Differentiation of macrophages in the microenvironment is remarkably dynamic since macrophages can quickly transition from one phenotype to the other based on the microenvironment or stimulation [31]. M1/M2 macrophage balance polarization determines the inflammatory status and homeostasis; M1 is considered to be pro-inflammatory, while M2 macrophages suppress inflammation by secreting high amounts of interleukin 10 (IL-10) and transforming growth factor-beta (TGF-β) [32].

In our study, another major change in immune cells by verteporfin treatment is TAM-M1 and M2. Our flow cytometry data showed an increased M1/M2 ratio (CD80/CD163 and CD86/CD163), which was different from scRNA-seq data that showed a significantly increased TAM-M1 population with verteporfin treatment. Although there is a discrepancy between these two sets of data in terms of cell composition, our data showed an overall decrease in the tumor volume with verteporfin treatment, which matched either the increased overall ratio of M1/M2 by flow cytometry or increased the TAM-M1 population by scRNA-seq.

Verteporfin was reported previously to decrease the stem cell marker Oct4, protein expression of epithelial–mesenchymal transition marker N-cadherin and spheroid formation [18]. YAP upregulates cancer stemness properties and phenotypes via Sox9, and verteporfin inhibits those characteristics [33]. We tested verteporfin using the animal model derived by YAP/Akt transduction and investigated if verteporfin affects cancer stemness. Our single-cell analysis showed that malignant cells with the expression of cancer stemness genes were reduced in the verteporfin-treated group. This result suggests that verteporfin reduced tumorigenesis partially derived by the inhibition of cancer stemness.

Recent transcriptomics analysis of human CCA suggested that human iCCA can be classified into four different groups based on the stroma, immune and tumor microenvironment [34]. One of the groups, characterized as a hepatic stem-like group, presented with high TAM-M2 infiltration, enrichment of Hippo/YAP pathway and Notch pathways, indicating potential therapeutic targets of CCA. Our study suggests that targeting the YAP pathway may be a potential candidate for drug development in CCA with a stemness feature.

## 5. Conclusions

Our data suggested the anti-tumoral activity of verteporfin in a YAP/AKT CCA animal model. We demonstrated that verteporfin remodels the immune environment, which might mediate the antitumoral effect of verteporfin through the immune response. In addition, our single-cell analysis data showed that verteporfin reduced the cancer stemness gene expressing malignant cells, suggesting that the inhibition of cancer stemness might also mediate tumor suppressive effect of verteporfin.

## Figures and Tables

**Figure 1 cancers-15-02454-f001:**
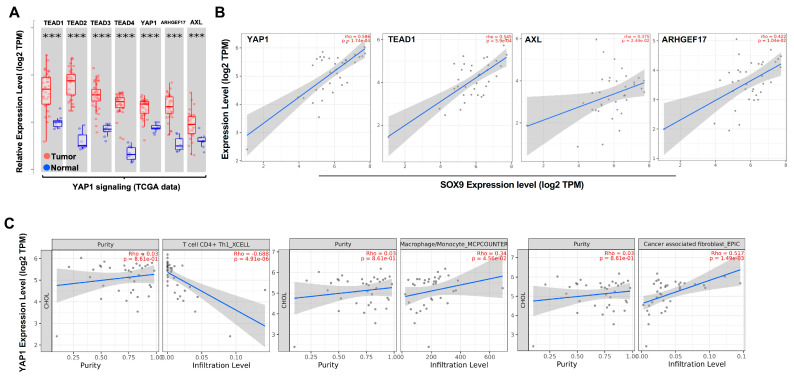
Analysis of correlation of expression levels of YAP signaling and stemness/immune cells in CCA using TCGA analysis. Each dot represents the informmation from a patient. (**A**) TIMER2.0 showed the expression of TEADs, YAP1 and target genes in the CCA cohort from the TCGA dataset and the corresponding normal tissues. (**B**) correlation between genes of YAP1 signaling and cancer stemness critical transcription factor SOX9. (**C**) Correlation between YAP1 expression with immune cell infiltration. ***: *p*-value < 0.001.

**Figure 2 cancers-15-02454-f002:**
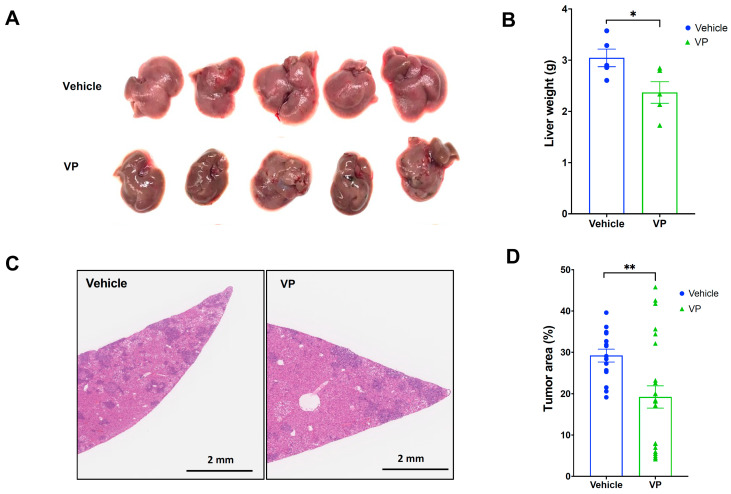
Verteporfin reduced liver weight and tumor area in YAP/AKT mouse CCA model. C57BL/6J mice were split into two groups (*n* = 5/each group), vehicle or verteporfin (VP), treated every three days starting at three weeks after YAP/Akt plasmid injection. (**A**) Images of livers. (**B**) Comparison of liver weight. (**C**) Representative images of H&E staining of livers. (**D**) Comparison of average tumor area of livers (three slides/samples). Values are means ± SD. * *p* < 0.05. ** *p* < 0.01. Veh, vehicle; VP, verteporfin.

**Figure 3 cancers-15-02454-f003:**
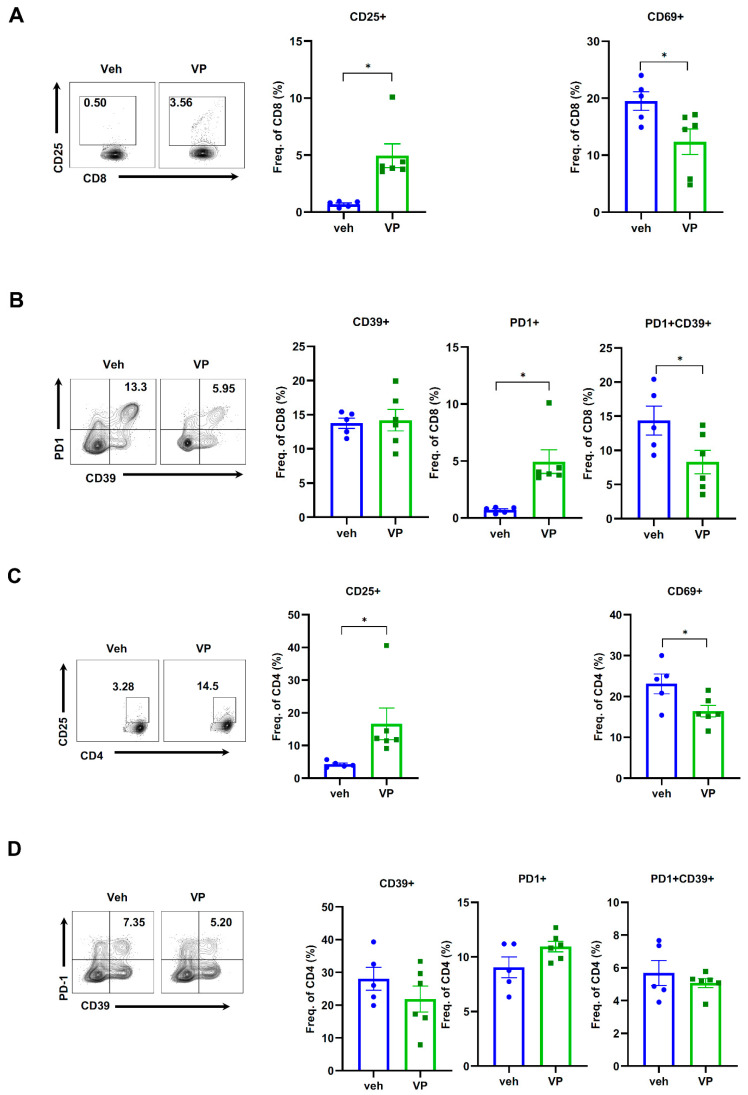
Verteporfin modulates T cell activation and exhaustion in YAP/AKT mouse CCA model. C57BL/6J mice were assigned into two groups (*n* = 6/each group), vehicle or verteporfin (VP), treated every three days three weeks after YAP/AKT plasmid injection. Liver samples were harvested and processed for flow cytometry analysis. (**A**) Frequency of activated CD8 T cells and (**B**) exhausted CD8 T cells among CD8 T cells. (**C**) Frequency of activated CD4 T cells and (**D**) exhausted CD4 T cells among CD4 T cells. Values are means ± SD. * *p* < 0.05. Veh, vehicle; VP, verteporfin.

**Figure 4 cancers-15-02454-f004:**
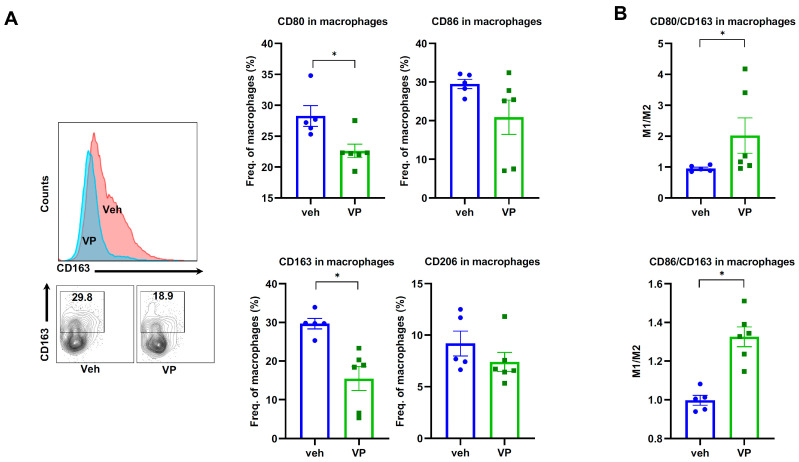
Verteporfin induces M2 to M1 macrophage transition in YAP/AKT mouse CCA model. C57BL/6J mice were split into two groups (*n* = 6/each group), vehicle or verteporfin (VP), treated every three days three weeks after YAP/AKT plasmid injection. Liver samples were harvested and processed for flow cytometry analysis of CD80, CD86, CD163 and ratio of M1/M2 (**A**) in CD11b-F4/80+ cells and (**B**) macrophages. Values are means ± SD. * *p* < 0.05. Veh, vehicle; VP, verteporfin.

**Figure 5 cancers-15-02454-f005:**
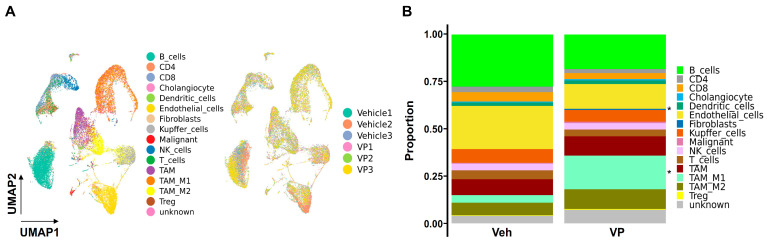
Single-cell-analysis-characterized immune cell population in YAP/AKT mouse model treated with verteporfin. C57BL/6J mice were split into two groups (*n* = 3 each group), vehicle or verteporfin (VP), treated every three days three weeks after YAP/AKT plasmid injection. A total of 33,769 cells were analyzed. (**A**) Scatter plot (UMAP) of scRNA-seq data with all cell types. (**B**) Bar plots of proportions of each cell type in vehicle and verteporfin treated group. * *p* < 0.05. Veh, vehicle; VP, verteporfin.

**Figure 6 cancers-15-02454-f006:**
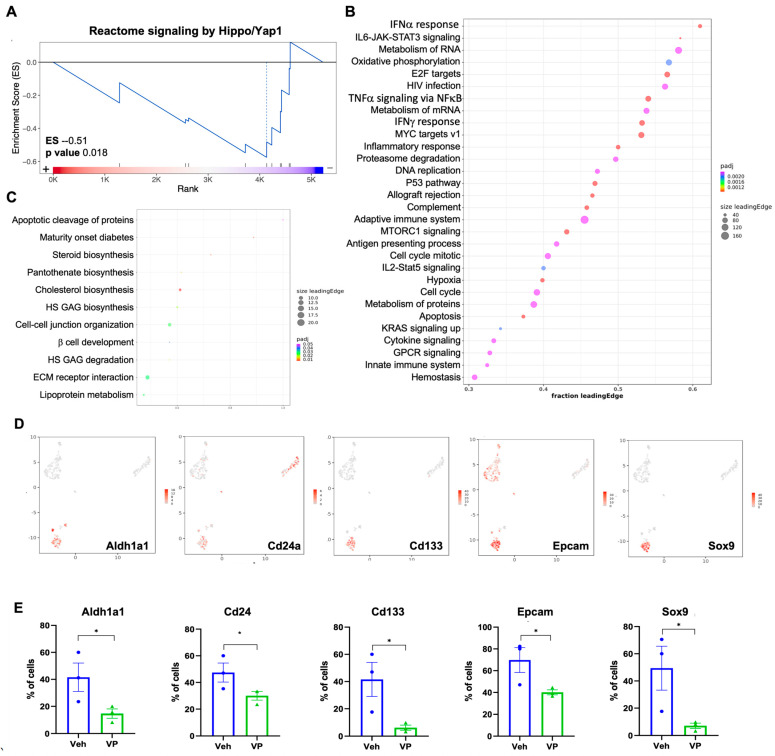
Single cell analysis showed that verteporfin modulates cancer stemness of YAP/AKT CCA model in malignant cells. C57BL/6J mice (*n* = 3 each group, for single cell analysis) were assigned into two groups, vehicle or verteporfin (VP) was treated every three days three weeks after YAP/Akt plasmids injection. Tumor/liver samples were dissociated and submitted for single-cell RNA sequencing. (**A**) GSEA analysis of all differentially expressed genes in all cells showed overall suppression of Hippo/YAP pathway with verteporfin treatment. (**B**) GSEA analysis of upregulated genes in malignant cells treated with verteporfin in comparison to the one treated with vehicle. (**C**) GSEA analysis of downregulated genes in malignant cells treated with verteporfin in comparison to the one treated with vehicle. (**D**) Markers for stemness cells and (**E**) changes in cancer stemness malignant cells by VP treatment in single cell analysis. * *p* < 0.05. Veh, vehicle; VP, verteporfin.

## Data Availability

The processed scRNA-seq dataset was deposited at the NCBI’s Gene expression omnibus (GEO) data repository. The remaining data are present in the article, Appendix A, or available from the authors upon reasonable request.

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
