# Peer review of "Anti-Cancer Activity of Verteporfin in Cholangiocarcinoma"

_cancers, 2023, doi:10.3390/cancers15092454_

Round 1

Reviewer 1 Report

In this manuscript, Golino and colleagues evaluated the effect of the treatment with verteporfin (VP), in mice in which was induced the formation of Cholangiocarcinoma (CCA) using hydrodynamic tail vein injection of plasmidic vectors carrying YAP, AKT and HSB2. This is a well-known methodology to generate an intriguing for of intrahepatic CCA origination mainly from neoplastic transformation of hepatocytes as demonstrated by the Calvisi group and by others.

In this manuscript, the Authors used primarily an in vivo approach and cutting edge technologies such as single cell analysis, supported by bioinformatics.

The most important results reached by the Authors are that A) VP-treated mice display a lighter liver with reduced tumor areas as respect to untreated controls. B) VP treatment induced the increased of M1 macrophage and CD8+ T cell populations. C) Treatment with VP is also able to reduce the number of stem-like cells as respect to unchallenged mice.

The paper is very interesting and could potentially pave the way for novel approaches for the treatment of iCCA burdened by the deregulation of Hippo pathway. The manuscript is clearly written, the experimental plan is adequate and the techniques are up to date. Few more experiments will strengthen the relevance of this excellent manuscript.

1.    Is very difficult to evaluate the morphology of the CCA generated by the hydrodynamic treatment, a higher magnification H&E micrographs added to figure 1 will be of great help for the readers. Moreover, the reviewer suggest to perform K19 staining to better highlight the tumoral cholagiocytes and to evaluate dysplastic areas.

2.    To reinforce data from FACS analysis, the reviewer suggests to perform immunohistochemistry/immunofluorescence staining and eventually quantification for the different population found differently regulated by VP treatment.

Author Response

Dear reviewer ,

Thank you all for your thoughtful and constructive comments. Essential experiments were conducted, and substantial modifications of manuscript were made to the body of the manuscript as you have advised. Below you will find an annotated list of comments and the revisions made.

1.  Is very difficult to evaluate the morphology of the CCA generated by the hydrodynamic treatment, a higher magnification H&E micrographs added to figure 1 will be of great help for the readers. Moreover, the reviewer suggest to perform K19 staining to better highlight the tumoral cholagiocytes and to evaluate dysplastic areas.

Answers: Thank you for your advice. We have included a higher magnification of H&E staining in the Figure 2C. We have performed CK19 IHC staining shown as following. We do not think this result is relevant to this manuscript (brown color indicates positive for CK19 staining). Therefore, we did not place into the manuscript.

2. To reinforce data from FACS analysis, the reviewer suggests to perform immunohistochemistry/immunofluorescence staining and eventually quantification for the different population found differently regulated by VP treatment.

Answers: Thank you for expertise input. FACS analysis has been a well-accepted method for immune cell profile given established antibody clones, which is much more reliable and high resolution in comparison to IHC. Multiplex immunofluorescence is also a relative high-resolution platform to direct visualize the immune cell composition and compare the change of immune cell profile with the treatment. However, it would take much longer time to develop the panels that work well on the samples we collected. Instead, we used our single cell RNA sequencing data that we generated in this study to further analyze immune cell composition change induced by verteporfin (Supplemental Fig. S6), though we understand that this platform is based on transcriptomic level while FACS or multiplex IF based on protein expression.

From the results shown here, only the change of percentage of CD80+ TAMs (TAM-M1) among TAMs significantly increased after verteporfin treatment. Other change showed similar tendency as FACS results though not reaching significant, for example, CD25+CD8+ and PD1+CD39+CD8+ cell populations. As we state above, scRNAseq data focuses on transcriptomic data while FACS presents protein expression data in immune cells. In addition, FACS analysis of immune cells in this manuscript only used isolated immune cells, while scRNAseq data was derived from whole cell population without specific enrichment process. It will certainly need further investigation in the future.

Reviewer 2 Report

The authors evaluated the verteporfin efficacy in the YAP murine model of cholangiocarcinoma (CCA). They performed sc-RNAseq and flow cytometry to identify the alterations related to cancer stemness and tumor immune infiltration of the CCA.

First of all, I think that the introduction should be expanded to provide a better understanding of the background. If this study is a continuum of the previous research some major findings or a view should be presented.

The description of the bioinformatic methods should be enriched with more details enabling the reproduction of the analyses. The GEO accession number to the data was not provided, but it should be. Heatmaps presenting the changes in the genes associated with mentioned pathway could be demonstrated and analyzed in greater detail, furthermore, these changes should be discussed more extensively. Some minor mistakes are to be corrected and the formatting, especially the last lines of the references.

Author Response

Dear reviewer,

Thank you all for your thoughtful and constructive comments. Essential experiments were conducted, and substantial modifications of manuscript were made to the body of the manuscript as you have advised. Below you will find an annotated list of comments and the revisions made.

1.  First of all, I think that the introduction should be expanded to provide a better understanding of the background. If this study is a continuum of the previous research some major findings or a view should be presented.

Answers: Thank you for the advice. We have extended the description in the introduction section.

  1. The description of the bioinformatic methods should be enriched with more details enabling the reproduction of the analyses.

Answers: Thank you for the suggestions. We adjust the flow of description and all analysis done were standard without special modification and all the codes used is deposited in GitHub in order to reproduce the results with the raw data provided.

  1. The GEO accession number to the data was not provided, but it should be.

Answers: Sorry for the unintentional neglection. GEO accession number is provided now.

  1. Heatmaps presenting the changes in the genes associated with mentioned pathway could be demonstrated and analyzed in greater detail, furthermore, these changes should be discussed more extensively.

Answers: Thank you for your advice. We have added more GSEA analysis results in the supplemental Fig. S7 and expanded discussion in this regard.

  1. Some minor mistakes are to be corrected and the formatting, especially the last lines of the references.

Answers: Thank you for checking. We have corrected those.

Round 2

Reviewer 1 Report

I believe that the immunohistochemical photo could have been included in the manuscript as a supplementary figure, but it is not such a serious concern as to affect the quality of the paper. As regards the immunohistochemical evaluation of the TRS cell populations, given the short time frame for the response that has been assigned, I understand that it was not possible to set the techniques and evaluate the sections